# Contribution of sociodemographic determinants in explaining the nutritional gap between the richest-poorest women of Bangladesh: A decomposition approach

Md. Sohel Rana[1]*, Md. Mobarak Hossain Khan[2]

**1** Department of Statistics, Comilla University, Kotbari, Bangladesh, **2** Department of Social Relations, East West University, Aftabnagar, Dhaka, Bangladesh

* sohel.573@gmail.com

**Data Availability Statement:** Data are available at http://dhsprogram.com/data/available-datasets.cfm.

## Abstract

### Background

Malnutrition among women disproportionately exists across socioeconomic classes of Bangladesh. According to our knowledge, studies which attempted to identify determinants and their contributions to explain BMI-based nutritional gap between the poorest and the richest categories of Wealth Index are still scarce.

### Objectives

To identify the nutritional gap of women between the richest-poorest classes in Bangladesh, and to determine how much of this gap are attributed to differences in predictors and differences in coefficients.

### Study population

Reproductive-aged (15–49 years) women of Bangladesh.

### Methods and procedures

We utilized the latest round (2017–2018) data of the Bangladesh Demographic and Health Survey (BDHS). Body mass index (BMI) has been used to measure the nutritional status of women. The kernel density was used to visualize the nutritional gap. The Oaxaca-Blinder (OB) decomposition method was used to ascertain influential determinants and their contributions to the existing gap between the richest-poorest classes of women.

### Results

We analyzed the data of 18,682 reproductive-aged women. There was a significant mean BMI gap of 4.1 unit (95% CI: 3.90–4.35) between the poorest-richest (25.6 vs 21.5) women. The overall prevalence of underweight, overweight and obese were 11.8%, 33.8% and 15.4%, respectively. The richest women were less underweight (7.5%) but more overweight

**Funding:** The authors received no specific funding for this work.

**Competing interests:** The authors have declared that no competing interests exist.

(23.7%) and obese (42.2%). In contrast, the poorest women were more underweight (32.0%) but less overweight (13.9%) and obese (7.0%). According to results of OB decomposition method, all predictors combinedly can explain 1.62 units (95% CI: 1.31–1.93) of the total mean BMI gap (equivalent to 40%). Some of the major predictors were women years of education (0.45 units, 95% CI: 0.27–0.64), spouse years of education (0.16 units, 95% CI: -0.02–0.34), current working status (0.17 units, 95% CI: 0.10–0.34), access to Television (0.50 units, 95% CI: 0.28–0.72), and place of residence (0.37 units, 95% CI: 0.22–0.72). The unexplained part of the poorest-richest gap was 2.51 units (95% CI: 2.13–2.89), which means that this particular gap will remain unchanged even though the mean difference of the predictors was diminished.

## Conclusions

A large part of the nutritional gap (approximately 60%) between the poorest and richest classes of women are found to be unchanged by the predictors of the study. Therefore, further predictors should be identified to minimize such gap. Moreover, policy makers and relevant stakeholders should implement feasible strategies to minimize the existing differences in the major predictors.

## 1. Background

Malnutrition is a worldwide public health problem. A few decades ago, undernutrition (underweight) was the common problem only for low and middle income countries (LMICs), whereas over-nutrition (overweight and obesity) was a matter of concern only for developed countries. Now-a-days, LMICs are facing both undernutrition and over-nutrition (called double burden of malnutrition) problems [1, 2]. Different studies revealed that both women and under-five children of Southeast Asia are at high-risk of double burden of malnutrition [1, 3–5]. A systematic review estimated the pooled prevalence of underweight and overweight for the period of 1969–2017, which were 28% and 17% respectively in the South Asian region. Another study reported the same prevalence (20%) of underweight and overweight in the Southeast Asia region [6]. According to the study of Mendez et al. (2005), the prevalence of overweight among the reproductive-aged women is higher than the prevalence of underweight in most of the developing countries [7].

Both forms of malnutrition are a major public health problem for children, adolescents and reproductive-aged women. The World Health Organization (WHO) reported that there were 1.9 billion adults who had been suffering from overweight or obesity in 2016, while the number for the underweight was about 462 million. From 1975 to 2016 the global prevalence of obesity became tripled. In 1975 just under 1% of children and adolescents were obese, but in 2016 the number was about 124 million (boys 8% and girls 6%) [8]. The Global status report on non-communicable diseases 2014 showed that the prevalence of overweight lies between 10% to 30% for most of the LMICs [9].

Many health problems are found to be associated with malnutrition. For instance, maternal underweight is associated with preterm birth and low birth weight [10, 11], malnourished children [12], and stillbirths. Moreover, undernutrition causes osteoporosis, asthma, anaemia, [13] mental health impairment [14], and increased risk of high mortality [15, 16]. Similarly, overweight is a significant risk factor for cardiovascular disease (CVD) [17–21], cancer,

diabetes mellitus [17–20, 22–25], hypertension [26, 27], stroke, respiratory problems [20, 23, 25], high cholesterol, high blood pressure, and arthritis [22, 23].

Malnutrition also impedes the productivity and reduces the gross domestic products (GDPs) and ultimately results a vicious cycle of poverty. The larger the burden of malnutrition, the larger the economic burden [28]. Underweight and overweight cost about 1% GDP in China and 2% GDP in India for treating non-communicable diseases [29]. It is projected that, by 2025, these expenditures could touch 9% in China [29]. Worldwide, hunger and undernutrition problems can reduce GDP by $1.4 to $2.1 trillion annually [28]. It is found that in an individual country overweight costs about 0.7–2.8% of the total expenditures in healthcare [30]. It is also estimated that underweight costs from 2.5% to 3.8% of a country's GDP and overweight causes 9.9% of overall healthcare costs in the Asia-Pacific region [31].

Rapid urbanization, demographic transition, and economic shift in the LMICs are the major causes of increasing prevalence of overweight along with the existing curse of underweight [1, 3, 4]. Overweight and obesity tend to have a positive association with various factors such as age, higher socioeconomic status, and urban residence [32–34]. Women from the richest households have more tendency to become overweight/obese, whereas women from the poorest households have more tendency to become underweight. Moreover, the likelihood of the prevalence of underweight among the urban women is less than rural counterparts, but the dimension is opposite for the prevalence of overweight-obesity [35]. Underweight is found to be more prevalent among young adult women (15–19 years) who belong to the poorest wealth quintile and who reside in rural region in south Asia and Southeast Asia [6]. In Bangladesh, household Wealth Index is positively associated with overweight-obesity, whilst, negatively related with underweight. The other determinants that are highly associated with malnutrition are higher parity and higher number of under five children at the household, lower educational level of women and their spouses, early age at first marriage, and food insecurity [35].

To the best of our knowledge, there is no study based on the 2017–2018 round of BDHS which aimed to estimate the overall and individual contribution of some selected socio-demographic predictors (factors) in explaining the nutritional gap between the poorest and richest group of reproductive aged women in Bangladesh. Therefore, following questions are addressed through this study: (1) What is the mean BMI gap between the poorest and richest groups (based on Wealth Index) of reproductive-aged women in Bangladesh? (2) How much of this gap can be attributed to the differences in selected predictors (called predictors effects)? (3) How much of the gap is attributed to the differences in the returns to the attributes (coefficient effects)? Such information is important for relevant policy makers and stakeholders for developing potential interventions.

## 2. Methods and procedures

### 2.1 Data source and study design

We extracted data from the women recode [BDIR] file of the latest round (2017–2018) of Bangladesh Demographic and Health Surveys (BDHSs). BDHS was started in 1993 and the latest round of BDHS (2017–2018) is the eighth survey. It is executed by the National Institute of Population Research and Training (NIPORT) of the Ministry of Health and Family Welfare (MOHFW) in collaboration with ICF, USA and financed by the United States Agency for International Development (USAID), Bangladesh. A Bangladeshi research farm named Mitra and Associates conducted the survey and ICF of Rockville, Maryland, USA, provided technical support. In 2017–2018 round of BDHS a two-stage stratified random sampling method has followed. The enumeration areas (EAs) from the 2011 National Population Housing Census (NPHC) by the Bangladesh Bureau of Statistics (BBS) has been used as a sampling frame. Each

EA is composed of 120 households on an average. As the primary sampling units (PSUs) in the first stage, 675 EAs (250 from urban areas and 425 from rural areas) were selected with probability sampling proportional to EA size. In the second stage, a systematic sample of 30 households on an average was taken from each EA. Thus, a total of 20,250 households were selected, of which 20,127 ever-married reproductive-aged women were interviewed. The more details about the sampling design, questionnaire, data collection and data processing are available elsewhere [36].

## 2.2 Outcome variables

We have considered BMI as an outcome variable, which is based on height and weight. BDHS considers it as a measure of nutritional status of ever-married women of age 15–49 years. BMI is defined as the ratio of weight in kilogram (kg) to the height in meter squared ($m^2$). BMI has been categorized to show the prevalence of underweight, overweight and obesity of the study population. Although, WHO suggested seven threshold of BMI for Asian countries [37], it is evident that 23.00 kg/$m^2$ and more BMI for the Asian people are considered as the risk of obesity [37]. Also, according to earlier studies conducted in India [38] and Bangladesh [35], individuals with BMIs ranging from 23.00 to 24.99 kg/$m^2$ are characterized as "at-risk for overweightness". That is why the cut-offs for the Asian countries has been followed in our study as a measure of malnutrition: <18.50 kg/$m^2$ (underweight), 18.50–22.99 kg/$m^2$ (normal weight), 23.00–27.49 kg/$m^2$ (overweight), and ≥27.5 kg/$m^2$ (obese).

## 2.3 Selected predictors

We have selected several sociodemographic and socioeconomic predictors based on contemporary literature review, availability of data and appropriateness. The current age of woman in years [6, 35], wealth index (poorest, poorer, middle, richer, richest), region (Barisal, Chittagong, Dhaka, Khulna, Mymensingh, Rajshahi, Rangpur, and Sylhet) [39], place of residence (urban, rural) [6, 35], educational attainment in years [6], spouse years of education, the number of children ever born, current working status, the status of contraception use, age at first marriage in years, current marital status, and frequency of TV watching are used as predictors in this study.

## 2.4 Statistical analysis

The major objective of this study was to determine the contribution of socio-demographic predictors to explain the nutritional gap between the poorest-richest reproductive aged women in Bangladesh. The kernel density was used to show the poorest-richest BMI gap. Different summary statistics are also presented in harmony with the kernel density. In order to determine the prevalence of malnutrition as well as its association with other sociodemographic predictors, the Asia specific cut-offs of BMI have been used. However, the continuous measure of BMI was used to find the contribution of each predictor to explain the nutrition gap between the poorest-richest women. The Oaxaca-Blinder (OB) decomposition [40, 41] permits us to divide the BMI difference by poorest and richest groups into two separate parts namely the "explained" part which measures the mean difference attributed to the predictors, and the "unexplained" part which represents the differences in the coefficient estimates. The OB method was originated by economists to measure the wage differentials between male and female. Now-a-days this method is used to explore the inequality in the health sectors. For detailed decomposition we used the OB method which provided the individual contribution of the predictors in explaining the richest-poorest difference in women's BMI. The OB method also allows us to report the contribution of unobserved variables explaining the poorest-richest

gap of BMI which cannot be reported by the typical regression-based methods [42]. We included "wealth" as an additional indicator predictor in the pooled regression to avoid the "index number problem," i.e. distortion due to residual group difference following the suggestion of Jann (2008). The elaborate procedures of the OB decomposition will be found elsewhere [42]. For model selection we utilized the forward selection criteria. We initially started with the model containing no predictors (provides only the intercept), then added the most significant predictors one by one. We also included the predictors having the smallest p-value, where we considered 0.05 as the threshold for statistically significant p-value. Akaike information criterion (AIC) is used in determining the best fitted model. STATA software (version 16.0) and Microsoft Excel has been used to analyse the data. To comply with the complex survey design of BDHS proper weighting has been done before analysing the data.

## 3. Results

We analyzed the data of 18,682 reproductive-aged women with complete data. Other cases are deleted due to incomplete data. Fig 1 represents the proportion of reproductive-aged women of four category of BMI according to their wealth index status. Women belonging to the poorest wealth quintile are less likely to be overweight-obese than the women belonging to richer and the richest quintiles. The figure reveals that the prevalence of overweight-obese women has increased and underweight and normal weight has decreased from the poorest to the richest quintiles of wealth index. Fig 2 reveals that there is a difference in the intersection point from the top of the *kernel density* of BMI of the poorest-richest women of reproductive age, which is a measure of the mean BMI difference between the poorest-richest women. Moreover, the long right tails of the kernel density of the richest women than the poorest women depict higher prevalence of overweight-obese falling in the richest wealth quintile.

Table 1 presents descriptive information of BMI and other background characteristics (e.g., age, division, women education in years) of the reproductive-aged women by total sample and by individual quintile of the wealth index. The mean BMI for the total women is 23.3 (SE: 0.05), which is 21.5 for the poorest wealth quintile and 25.6 for the richest wealth quintile. Hence the gap of mean BMI between the poorest-richest quintiles of the reproductive aged women is 4.1. The prevalence of underweight is approximately five times higher among the women living in the poorest wealth quintile than the richest ones (20.3% vs 4.3%). However, we observe opposite tendency for overweight and obese. The prevalence of overweight women in the poorest and the richest wealth quintile is 25.2% and 39.3% respectively.

The proportion of obese women in the richest quintile is about 5.5 times higher than the poorest quintile (32.0% vs 5.8%). Age distribution shows that more than two-third (66.8%) of the women belongs to the age group of 20–39 years. Form the eight division of Bangladesh, Dhaka division contributes one-fourth of the total sample. Mean years of woman education for the total sample is about 5.5 years, although it was 8.2 years for the richest quintile and 3.3 years for the poorest quintile. Similarly, the percentage of women having no formal education falling in the poorest wealth quintile is approximately four times higher than the richest quintile (34.9% vs 8.1%). The rate of higher education (i.e., more than 12 years) is far advanced for the richest than the poorest quintiles (10.7% vs 0.20%). Almost similar pattern is observed for the spouse education. Women living in the poorest wealth quintile gave more birth than their richest counterparts. For instance, the percentage of women with more than three children is 50.8% in the poorest and 32.1% in the richest group. A higher percentage of the poorest women currently engaged in works than the richest women (63.1% vs 30.6%). The percentage of women who watch TV at least once a week is six times higher among the richest than the poorest quintiles (86.3% vs 14.1%).

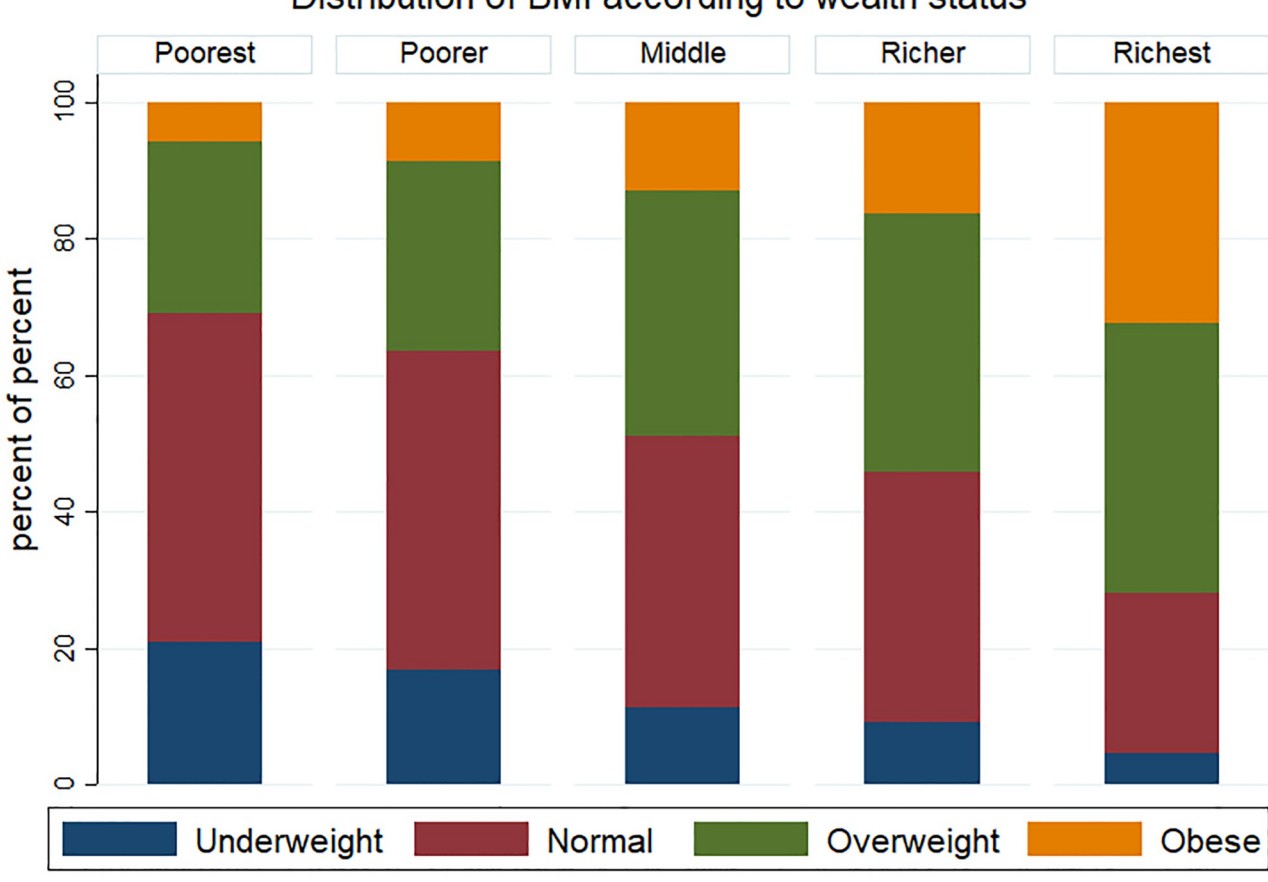

**Fig 1. Distribution of body mass index (BMI) according to the wealth index of reproductive-aged (15–49 years) women of Bangladesh.**

Table 2 presents the bivariate association between wealth index and socio-demographic factors of reproductive-aged women. The results reveal that the richest women are less likely to be underweight and more likely to be overweight and obese (7.5% vs 23.7% vs 42.2%). On the other hand, women living in the poorest wealth quintile are more likely to be underweight and less likely to be overweight and obese (32.0% vs 13.9% vs 7.0%). In urban area, the percentage of the richest women is higher than the poorest (46.4% vs 6.7%), whereas almost half of the rural women are poorer and poorest. Table 2 also discloses that the richest the women and their spouse the more educated the women and their partner. The richest women have more access to watch television than the poorest women. Usually, richest women got married at older age than the poorest women.

The results of OB decomposition analysis, which is performed to estimate the contribution of socio-demographic predictors in explaining the mean BMI gap (total predicted gap) of 4.1 (95% CI: 3.90–4.35) between the poorest-richest women, is presented in Table 3. The intercept terms for the regression model of the richest, poorest and the pooled groups, which represent the average BMI for these groups of women are 19.31, 19.09 and 17.05 respectively, when all other predictors remain constant. Moreover, all these intercepts are statistically significant at 0.1% level. However, the difference of the intercepts as part of the unexplained portion is not statistically significant. The predictors included in the OB decomposition can explain 1.62 units (95% CI: 1.31–1.93) of the total mean BMI difference between the richest-poorest

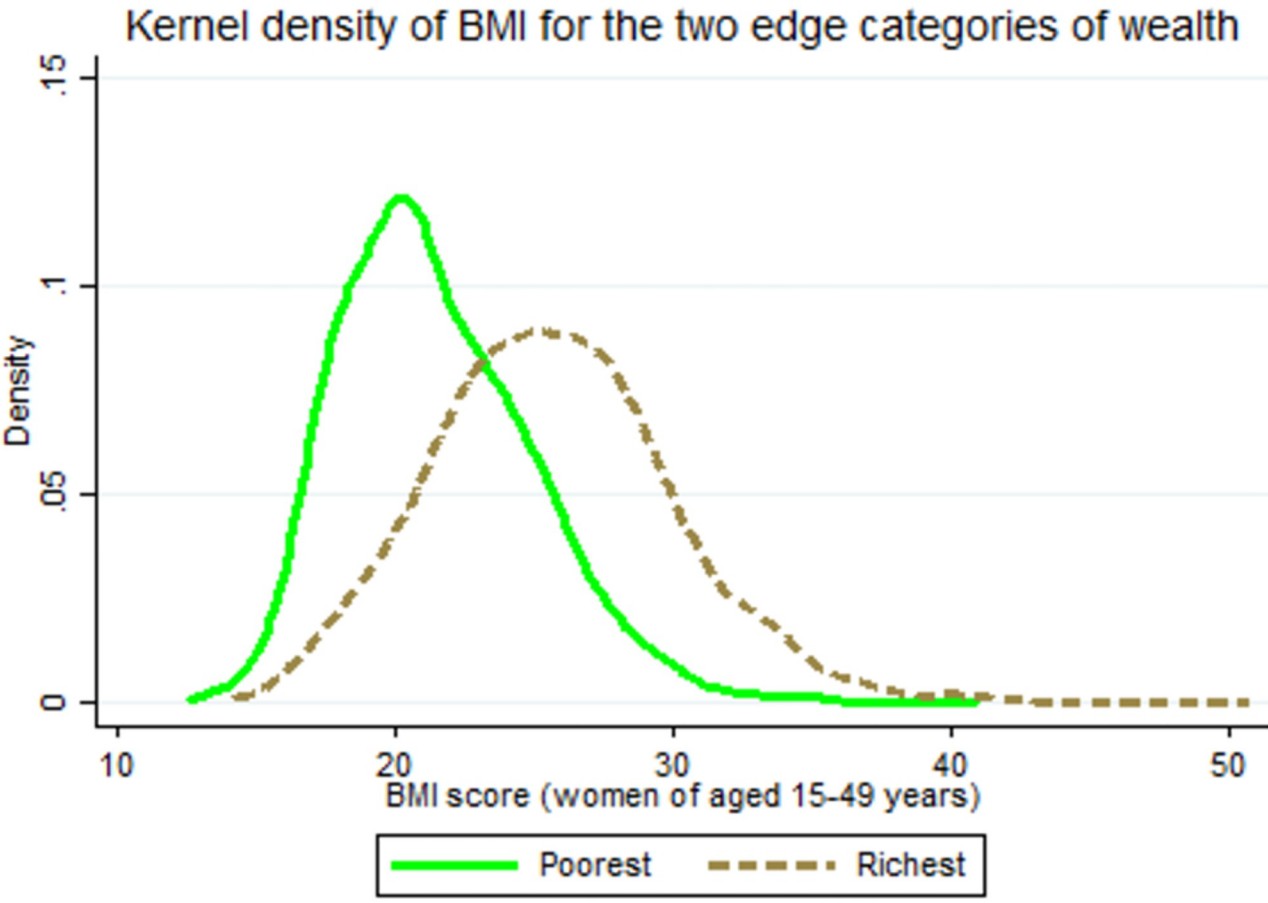

**Fig 2. Distribution of body mass index (BMI) of the poorest-richest group of reproductive aged women of Bangladesh.**

women of Bangladesh. The major predictors in explaining the mentioned BMI difference are women years of education (0.45 units, 95% CI: 0.27–0.64), spouse years of education (0.16 units, 95% CI: -0.02–0.34), current working status (0.17 units 95% CI: 0.10–0.34), access of television (0.50 units, 95% CI: 0.28–0.72), and place of residence (0.37 units, 95% CI: 0.22–0.72).

The explained portion (= 1.62) means that the difference of mean BMI between the richest-poorest women is due to the difference in mean of the predictors included in the OB decomposition. This difference can be removed if the poorest and the richest women have the same characteristics in the predictors. The negative sign of predictors indicates the difference of mean BMI between the richest-poorest women will increase due to its effects instead of decrease. The value of age at first marriage -0.05 units (95% CI: -0.11–0.01) means the mean difference of BMI between the richest-poorest women will increase instead of decrease if the age at first marriage is same for both groups. Similar interpretation is applicable for use of contraception (-0.02 units, 95% CI: -0.04–0.00) and total children ever born (-0.04 units, -0.10–0.03). The unexplained portion (= 2.51 units, 95% CI: 2.13–2.89) specifies that the total mean BMI difference between the richest-poorest women remain same even though the mean difference of the predictors remain same for both groups. This difference is due to the difference in the coefficient estimates. The unexplained part happen due to three main reasons namely omitting of influential predictors, measurement errors between the group, and discrimination

**Table 1. Descriptive information of BMI and background characteristics by total sample and by individual quintile of the wealth index, BDHS 2017–18.**

| Variable | Whole (18682) | Poorest (3554) | Poorer (3574) | Middle (3639) | Richer (3809) | Richest (4106) |
|---|---|---|---|---|---|---|
| | % (N) | % (N) | % (N) | % (N) | % (N) | % (N) |
| BMI [Mean (SE)]¶ | 23.3 (0.05) | 21.5 (0.08) | 22.2 (0.07) | 23.2 (0.07) | 23.8 (0.09) | 25.6 (0.09) |
| Malnutrition prevalence | | | | | | |
| Underweight | 11.8 (2285) | 20.3 (739) | 15.9 (603) | 10.5 (408) | 9.0 (349) | 4.3 (186) |
| Normal weight | 39.0 (7185) | 48.7 (1717) | 46.4 (1664) | 40.3 (1446) | 36.1 (1390) | 24.4 (968) |
| Overweight | 33.8 (6264) | 25.2 (895) | 28.4 (993) | 36.3 (1311) | 38.6 (1443) | 39.3 (1622) |
| Obese | 15.4 (2948) | 5.8 (203) | 9.2 (314) | 13.0 (474) | 16.3 (627) | 32.0 (1330) |
| Age [Mean (SE)]¶ | 31.9 (0.08) | 31.6 (0.17) | 32.1 (0.15) | 32.0 (0.16) | 31.4 (0.16) | 32.3 (0.17) |
| 15–19 Years of old | 9.1 (1610) | 9.7 (341) | 9.7 (360) | 10.0 (342) | 9.6 (340) | 6.5 (227) |
| 20–29 Years of old | 34.2 (6334) | 34.3 (1225) | 32.3 (1147) | 32.6 (1185) | 36.1 (1342) | 35.7 (1435) |
| 30–39 Years of old | 32.6 (6147) | 33.1 (1180) | 32.6 (1169) | 32.5 (1193) | 31.4 (1213) | 33.3 (1392) |
| 40–49 Years of old | 24.1 (4591) | 23.0 (808) | 25.3 (898) | 24.8 (919) | 22.8 (914) | 24.5 (1052) |
| Division | | | | | | |
| Barisal | 5.6 (2013) | 9.4 (560) | 6.7 (420) | 6.2 (461) | 3.9 (329) | 2.2 (243) |
| Chittagong | 17.8 (2670) | 13.1 (346) | 14.3 (378) | 20.1 (559) | 18.2 (566) | 23.0 (821) |
| Dhaka | 25.1 (2704) | 11.2 (194) | 15.9 (296) | 20.3 (404) | 34.0 (769) | 42.2 (1041) |
| Khulna | 11.8 (2478) | 7.5 (263) | 13.3 (486) | 14.6 (570) | 12.9 (594) | 10.2 (565) |
| Mymensingh | 7.7 (2002) | 12.0 (541) | 9.9 (491) | 7.3 (389) | 5.9 (336) | 3.7 (245) |
| Rajshahi | 14.2 (2433) | 15.4 (445) | 17.4 (538) | 16.7 (556) | 13.2 (506) | 8.4 (388) |
| Rangpur | 12.1 (2344) | 23.9 (795) | 16.3 (572) | 10.5 (400) | 6.6 (302) | 4.4 (275) |
| Sylhet | 5.8 (2038) | 7.5 (410) | 6.3 (393) | 4.3 (300) | 5.3 (407) | 5.9 (528) |
| Education in years[Mean (SE)]¶ | 5.5 (0.06) | 3.3 (0.07) | 4.2 (0.07) | 5.4 (0.08) | 6.2 (0.1) | 8.2 (0.12) |
| No formal education | 20.1 (3622) | 34.9 (1194) | 26.6 (934) | 17.8 (656) | 14.9 (554) | 8.1 (284) |
| 1–5 years | 31.9 (6002) | 40.8 (1500) | 38.4 (1400) | 33.8 (1247) | 29.0 (1130) | 18.7 (725) |
| 6–12 years | 44.7 (8332) | 24.1 (851) | 34.7 (1222) | 47.2 (1691) | 53.0 (2003) | 62.5 (2565) |
| More than 12 years | 3.2 (726) | 0.2 (9) | 0.4 (18) | 1.2 (45) | 3.1 (122) | 10.7 (532) |
| Spouse education in years [Mean (SE)]¶ | 5.3 (0.07) | 2.6 (0.07) | 3.7 (0.08) | 5.1 (0.1) | 6.1 (0.11) | 8.6 (0.14) |
| No formal education | 29.5 (5476) | 50.5 (1769) | 36.7 (1326) | 26.7 (993) | 22.3 (855) | 13.6 (533) |
| 1–5 years | 28.7 (5246) | 33.6 (1210) | 36.4 (1308) | 32.5 (1169) | 25.9 (971) | 16.0 (588) |
| 6–12 years | 35.1 (6550) | 15.5 (559) | 25.8 (896) | 36.8 (1335) | 44.6 (1703) | 50.8 (2057) |
| More than 12 years | 6.6 (1410) | 0.4 (16) | 1.2 (44) | 4.0 (142) | 7.1 (280) | 19.6 (928) |
| # children ever born [Mean (SE)]¶ | 2.5 (0.02) | 2.8 (0.04) | 2.7 (0.03) | 2.5 (0.03) | 2.3 (0.03) | 2.1 (0.03) |
| 0 | 8.2 (1505) | 5.8 (209) | 6.9 (249) | 7.4 (266) | 10.0 (349) | 10.7 (432) |
| 1–2 | 49.5 (9342) | 43.4 (1525) | 44.9 (1629) | 49.5 (1807) | 51.6 (1989) | 57.2 (2392) |
| > = 3 | 42.3 (7835) | 50.8 (1820) | 48.2 (1696) | 43.2 (1566) | 38.4 (1471) | 32.1 (1282) |
| Currently working | | | | | | |
| No | 51.2 (9533) | 36.9 (1321) | 40.6 (1450) | 51.0 (1775) | 56.2 (2128) | 69.4 (2859) |
| Yes | 48.8 (9149) | 63.1 (2233) | 59.4 (2124) | 49.0 (1864) | 43.8 (1681) | 30.6 (1247) |
| Using contraception | | | | | | |
| No | 38.1 (7024) | 34.5 (1234) | 36.2 (1276) | 40.5 (1442) | 38.2 (1435) | 40.7 (1637) |
| Yes | 61.9 (11658) | 65.5 (2320) | 63.8 (2298) | 59.5 (2197) | 61.8 (2374) | 59.3 (2469) |
| TV access | | | | | | |
| Not at all | 35.9 (6848) | 75.4 (2737) | 49.6 (1813) | 29.8 (1119) | 19.8 (797) | 9.2 (382) |
| Less than once a week | 9.2 (1651) | 10.5 (342) | 13.5 (469) | 10.0 (366) | 8.0 (292) | 4.4 (182) |
| At least once a week | 54.8 (10183) | 14.1 (475) | 36.9 (1292) | 60.2 (2154) | 72.2 (2720) | 86.3 (3542) |
| Age at 1st Marriage | | | | | | |
| < 18 years | 75.9 (13910) | 83.9 (2967) | 81.2 (2877) | 80.3 (2926) | 74.1 (2771) | 61.1 (2369) |

*(Continued)*

**Table 1.** (Continued)

| Variable | Whole (18682) | Poorest (3554) | Poorer (3574) | Middle (3639) | Richer (3809) | Richest (4106) |
|---|---|---|---|---|---|---|
| | % (N) | % (N) | % (N) | % (N) | % (N) | % (N) |
| > = 18years | 24.1 (4772) | 16.1 (587) | 18.8 (697) | 19.7 (713) | 25.9 (1038) | 38.9 (1737) |
| Partnership Status | | | | | | |
| Not living together | 6.0 (1207) | 7.6 (273) | 6.1 (232) | 5.5 (223) | 5.0 (223) | 5.9 (256) |
| Living together | 94.0 (17475) | 92.4 (3281) | 93.9 (3342) | 94.5 (3416) | 95.0 (3586) | 94.1 (3850) |

ꬶ indicates continuous variables.

% are weighted to account for survey design, but N is unweighted.

between the group. The R-squared value of 0.11, 0.03 and 0.26 for the regression model of women in the richest, poorest and the pooled model means the predictors included in the model can explain 11%, 3% and 26% of the total variation in mean BMI.

## 4. Discussion

In this study, we have examined the nutrition gap between the two extreme categories of wealth quintiles by measuring the difference of mean BMI and found that some determinants are causing (based on OB decomposition method) this difference. Though the mean BMI of the richest group of women revealed that they are residing in the overweight category if we consider the Asia-specific thresholds of BMI. However, the association between malnutrition and wealth quintiles show that most of the richest women are overweight and obese as well as the prevalence of underweight is also not negligible. Similarly, mean BMI of the poorest women depicts that they are belonging to normal weight category. But, the association between different categories of malnutrition and wealth quintiles exhibited that the prevalence of underweight and normal weight is dominating, yet overweight-obesity is coexisting. This trend coincides with the previous studies [6, 31, 33, 35].

Education for women and their spouses are playing a vital role in nutritional difference. The more the education the more likelihood of being overweight-obese. The possible reasons behind this matter are that educated peoples are habituated on sedentary life styles, and most of them are doing table-chair oriented work (i.e. passing more time in sitting) [32, 43, 44]. Another reason may be there is a trend of taking fast food and processed food among the educated women and men in Bangladesh. On the other hand, less educated people generally do manual work by which they spend lots of time in physically vigorous situation. Such vigorous activities may help less educated people to burn excess fat of the body. Moreover, illiterate or less educated peoples earn less money, as a result they fail to take nutritious food. These are also possible reasons for the higher prevalence of underweight among the poorest women of Bangladesh. Other studies have reported similar results [6, 33, 45]. Frequency of TV watching is also exhibited as an influential driver of being overweight and obese. This determinant actually measures the sitting time of an individual. The more is the sitting time, the less is physically active, and helps to grow the weight of Bangladeshi women. An adult should run or walk one hour per day to remain physically sound and healthy. If it is not possible for some adults, it is advisable to spend at least half an hour for physical exercise. Generally, for women who spend more time for watching TV or browsing smart phones are at increased risk of having higher BMI and poor physical and mental health [46, 47].

By place of residence, there exists a difference of mean BMI between the richest-poorest women of Bangladesh. Several previous studies [7, 19, 24, 48] showed that urbanization is a risk factor for overweightness and obesity. Our analysis has also found the similar results. May

**Table 2. Percentages of wealth index categories by various socio-demographic variables and their bi-variate associations among the reproductive aged women of BDHS 2017–18.**

| Variable | Poorest | Poorer | Middle | Rich | Richest |
|---|---|---|---|---|---|
| | % (95% CI) | % (95% CI) | % (95% CI) | % (95% CI) | % (95% CI) |
| Body mass index | | | | | |
| Underweight | 32.0 (29.0–35.2) | 26.6 (24.5–28.9) | 18.1 (16.1–20.1) | 15.8 (13.9–18.0) | 7.5 (6.1–9.1) |
| Normal weight | 23.3 (21.4–25.4) | 23.5 (22.1–25.0) | 21.0 (19.6–22.4) | 19.4 (17.9–20.9) | 12.8 (11.5–14.1) |
| Overweight | 13.9 (12.5–15.6) | 16.6 (15.3–18.0) | 21.8 (20.4–23.3) | 23.9 (22.3–25.5) | 23.7 (22.0–25.6) |
| Obese | 7.0 (5.9–8.2) | 11.7 (10.3–13.3) | 17.0 (15.4–18.8) | 22.0 (20–24.2) | 42.2 (39.5–45) |
| P-value | P<0.001 | | | | |
| Current Age | | | | | |
| 15–19 Years | 19.8 (17.1–22.8) | 21.1 (18.9–23.6) | 22.3 (19.9–25.0) | 22.1 (19.4–25) | 14.6 (12.4–17.2) |
| 20–29 Years | 18.7 (17.0–20.5) | 18.6 (17.3–20.0) | 19.4 (18.1–20.7) | 22.0 (20.4–23.8) | 21.3 (19.5–23.2) |
| 30–39 Years | 18.9 (17.3–20.7) | 19.8 (18.5–21.2) | 20.3 (18.9–21.8) | 20.1 (18.7–21.6) | 20.9 (19.2–22.6) |
| 40–49 Years | 17.8 (16.0–19.8) | 20.7 (19.2–22.3) | 20.9 (19.5–22.4) | 19.8 (18.3–21.4) | 20.7 (18.9–22.7) |
| P-value | P<0.001 | | | | |
| Place of Residence | | | | | |
| Urban | 6.7 (5.2–8.7) | 6.5 (5.4–7.7) | 12.0 (10.3–14) | 28.3 (25.6–31.3) | 46.4 (42.5–50.4) |
| Rural | 23.3 (21.3–25.5) | 24.9 (23.6–26.3) | 23.6 (22.3–24.8) | 18.0 (16.7–19.3) | 10.2 (9.0–11.5) |
| P-value | P<0.001 | | | | |
| Division | | | | | |
| Barisal | 31.4 (25.1–38.4) | 23.5 (20.8–26.5) | 22.5 (19.4–26.1) | 14.4 (12.0–17.2) | 8.2 (6.1–10.8) |
| Chittagong | 13.7 (10.0–18.5) | 15.8 (13.3–18.7) | 22.9 (20.2–25.8) | 21.3 (18.6–24.3) | 26.3 (22.9–30.0) |
| Dhaka | 8.4 (6.3–11.1) | 12.5 (10.3–15.2) | 16.4 (14.0–19.2) | 28.3 (24.9–31.9) | 34.4 (30.1–38.8) |
| Khulna | 11.8 (9.0–15.4) | 22.3 (19.7–25) | 25.3 (22.9–27.8) | 22.9 (20.4–25.8) | 17.7 (14.7–21.1) |
| Mymensingh | 29.2 (23.9–35.2) | 25.5 (21.9–29.4) | 19.3 (16.5–22.3) | 16.1 (12.4–20.8) | 9.9 (7.5–13.0) |
| Rajshahi | 20.3 (15.9–25.6) | 24.2 (21.8–26.9) | 24.0 (21.3–26.8) | 19.5 (16.8–22.4) | 12.1 (9.7–14.9) |
| Rangpur | 37.0 (31.9–42.3) | 26.6 (23.7–29.7) | 17.6 (15.3–20.2) | 11.4 (9.6–13.6) | 7.4 (5.5–9.8) |
| Sylhet | 23.9 (17.4–31.8) | 21.4 (18–25.2) | 15.1 (12.8–17.8) | 19.0 (15.9–22.4) | 20.7 (16.2–26) |
| P-value | P<0.001 | | | | |
| Women's education (yrs) education | | | | | |
| No formal education | 32.4 (29.6–35.3) | 26.1 (24.1–28.2) | 18.0 (16.3–19.8) | 15.5 (13.6–17.5) | 8.2 (6.8–9.7) |
| 1–5 years | 23.8 (21.9–25.9) | 23.7 (22.3–25.3) | 21.5 (20.0–23.1) | 19.0 (17.5–20.6) | 11.9 (10.7–13.4) |
| 6–12 years | 10.0 (9.0–11.2) | 15.3 (14.2–16.4) | 21.4 (20.2–22.7) | 24.7 (23.3–26.1) | 28.5 (26.6–30.5) |
| More than 12 years | 1.4 (0.7–2.8) | 2.3 (1.4–3.7.0) | 7.6 (5.6–10.3) | 20.1 (16.3–24.5) | 68.6 (63.2–73.6) |
| P-value | P<0.001 | | | | |
| Spouse Education | | | | | |
| No formal education | 31.9 (29.4–34.6) | 24.5 (22.9–26.2) | 18.3 (16.8–20.0) | 15.8 (14.2–17.5) | 9.4 (8.3–10.7) |
| 1–5 years | 21.8 (19.9–23.9) | 25.0 (23.4–26.7) | 23.0 (21.5–24.5) | 18.8 (17.3–20.5) | 11.3 (10.0–12.8) |
| 6–12 years | 8.2 (7.2–9.4) | 14.5 (13.2–15.8) | 21.3 (20.0–22.7) | 26.5 (24.9–28.2) | 29.5 (27.5–31.6) |
| More than 12 years | 1.1 (0.6–2.0) | 3.5 (2.5–4.8) | 12.4 (10.0–15.4) | 22.6 (19.6–25.8) | 60.4 (56.2–64.4) |
| P-value | P<0.001 | | | | |
| Parity | | | | | |
| 0 | 13.2 (11.0–15.7) | 16.6 (14.5–18.8) | 18.3 (16.0–20.7) | 25.5 (22.6–28.5) | 26.5 (23.5–29.8) |
| 1–2 | 16.4 (15.0–17.9) | 17.9 (16.8–19.1) | 20.3 (19.1–21.5) | 21.8 (20.5–23.2) | 23.6 (21.9–25.4) |
| > = 3 | 22.4 (20.4–24.5) | 22.5 (21.1–23.9) | 20.7 (19.4–22.1) | 18.9 (17.6–20.3) | 15.5 (14.1–17.0) |
| P-value | P<0.001 | | | | |
| Current working status | | | | | |
| No | 13.5 (12–15.1) | 15.7 (14.5–16.9) | 20.2 (18.9–21.7) | 22.9 (21.4–24.6) | 27.7 (25.6–29.9) |

*(Continued)*

**Table 2.** (Continued)

| Variable | Poorest | Poorer | Middle | Rich | Richest |
|---|---|---|---|---|---|
| | % (95% CI) | % (95% CI) | % (95% CI) | % (95% CI) | % (95% CI) |
| Yes | 24.1 (22.1–26.2) | 24.0 (22.7–25.3) | 20.4 (19.2–21.7) | 18.7 (17.3–20.2) | 12.8 (11.6–14.0) |
| P-value | P<0.001 | | | | |
| Contraception use status | | | | | |
| Not using | 16.9 (15.2–18.7) | 18.8 (17.4–20.2) | 21.6 (20.2–23.1) | 20.9 (19.5–22.5) | 21.8 (20.0–23.7) |
| Using | 19.7 (18.1–21.5) | 20.3 (19.2–21.5) | 19.5 (18.4–20.7) | 20.8 (19.5–22.2) | 19.5 (18.1–21.1) |
| P-value | P<0.001 | | | | |
| Age at first marriage | | | | | |
| Less than 18 years | 20.6 (18.9–22.4) | 21.1 (20.0–22.3) | 21.5 (20.4–22.7) | 20.4 (19.1–21.7) | 16.4 (15.1–17.8) |
| 18 years and more | 12.5 (10.7–14.6) | 15.4 (13.8–17.1) | 16.6 (15.2–18.1) | 22.5 (20.6–24.5) | 33.0 (30.4–35.7) |
| P-value | P<0.001 | | | | |
| Access to TV | | | | | |
| Not at all | 39.2 (36.4–42.0) | 27.3 (25.6–29.0) | 16.8 (15.5–18.3) | 11.5 (10.4–12.7) | 5.3 (4.5–6.1) |
| Less than once a week | 21.2 (18.9–23.8) | 28.8 (26.3–31.6) | 22.0 (19.7–24.5) | 18.1 (15.9–20.6) | 9.8 (8.1–11.7) |
| At least once a week | 4.8 (4.2–5.4) | 13.3 (12.3–14.4) | 22.3 (21–23.7) | 27.5 (25.9–29.1) | 32.1 (30.1–34.2) |
| P-value | P<0.001 | | | | |
| Marital Status | | | | | |
| Not living together | 23.6 (20.7–26.8) | 20.2 (17.7–22.9) | 18.6 (16.0–21.4) | 17.4 (15.0–20.2) | 20.2 (17.4–23.4) |
| Living together | 18.3 (16.8–20.0) | 19.7 (18.7–20.8) | 20.4 (19.4–21.5) | 21.1 (19.9–22.4) | 20.4 (19.0–21.9) |
| P-value | P<0.001 | | | | |

be lack of playground, park, footpath to walk, contaminated air in the urban region of Bangladesh are responsible for such tendency. Urban region is more furnished with the modern healthcare facilities. Moreover, more educated people living in urban region are habituated to eat processed foods due to widespread availability and higher purchasing power. On the other hand, rural people generally do more tedious job which requires more physical activity. The older the women the more likelihood of being overweight-obese [6, 16, 39, 49]. We also found that age is a factors for the nutrition gap between the poorest-richest women of Bangladesh. That means, the richest people are older than their poorest counterpart. But the effect of this determinants is not so strong.

The findings of our study suggest that to eliminate the nutritional gap between two extreme categories of wealth index, the mean difference in predictors should be reduced. We found that forty percent (1.62/4.13x100 = 40%) of the nutritional gap can be explained by the studied predictors. Almost 60% of the nutritional gap still remains unexplained, which may happen due to other reasons. Firstly, several factors related to BMI such as dietary intakes, physical activities, and intergenerational properties are not included in the OB decomposition due to unavailability of data in BDHS data sets. Secondly, errors in measurements describe the systematic distinguish of a factor by wealth. This type of error may exist among the predictors. For instance, within a category of socioeconomic status, the richest women may have higher levels compared to their poorest counterparts, indicating two similar things were not compared. Finally, the unexplained gap can be attributed to the discrimination between two groups, which means differences in access to food and healthcare based on their wealth index may influence their BMI [50]. As policy implications from the findings of our study, we would like to suggest relevant authorities to offer more attention to minimize the inequality of education for both reproductive aged women and their partners, mass media access, health facilities between urban-rural residence, physically active working environment. Moreover,

**Table 3. Sample means, regression estimates, and decomposition results for the richest-poorest group of women's.**

| Variable | Mean Richest | Mean Poorest | Coeff. Richest | Coeff. Poorest | Coeff. pooled | OB decomposition | |
|---|---|---|---|---|---|---|---|
| | | | | | | Explained (95% CI) | Unexplained (95% CI) |
| Current age (years) | 32.28 | 31.65 | 0.13*** | 0.07*** | 0.11*** | 0.07 (0.02; 0.12)*** | 1.98 (1.01; 2.94)*** |
| Women's education (years) | 8.18 | 3.34 | 0.09** | 0.06* | 0.09*** | 0.45 (0.27; 0.64)*** | 0.08 (-0.33; 0.49) |
| Spouse education (years) | 8.58 | 2.56 | 0.04 | 0.03 | 0.03 | 0.16 (-0.02; 0.34)* | 0.11 (-0.21; 0.43) |
| Currently working (yes/no)[a] | 0.31 | 0.63 | -0.83*** | -0.13 | -0.53*** | 0.17 (0.10; 0.24)*** | -0.34 (-0.55; -0.14)*** |
| Contraception (yes/no)[b] | 0.59 | 0.66 | -0.01 | 0.67*** | 0.35** | -0.02 (-0.04; 0.00)** | -0.43 (-0.71; -0.14)*** |
| TV access[c] | 0.86 | 0.14 | 0.84*** | 0.53* | 0.69*** | 0.50 (0.28; 0.72)*** | 0.15 (-0.14; 0.44) |
| Current partnership status[d] | 0.94 | 0.92 | 0.32 | 0.56* | 0.63** | 0.01 (0.00; 0.02)* | -0.23 (-1.14; 0.69) |
| Place of residence[e] | 0.36 | 0.90 | -0.72*** | -0.76*** | -0.69*** | 0.37 (0.22; 0.52)*** | 0.05 (-0.36; 0.45) |
| Age at 1st Marriage[f] | 0.39 | 0.16 | -0.03 | -0.32* | -0.23 | -0.05 (-0.11; 0.01)* | 0.09 (-0.03; 0.22) |
| Total children ever born | 2.06 | 2.81 | 0.31*** | -0.06 | 0.05 | -0.04 (-0.10; 0.03) | 0.83 (0.38; 1.27)*** |
| Wealth[e] | | | | | 0.63*** | | |
| Intercept | | | 19.31*** | 19.09*** | 17.05*** | | 0.22 (-1.16; 1.60) |
| Predicted BMI | 25.65 | 21.52 | | | | | |
| Total contribution | | | | | | 1.62 (1.31; 1.93)*** | 2.51 (2.13; 2.89)*** |
| Total predicted gap | | | | | | | 4.13 (3.90; 4.35)*** |
| R-squared | | | 0.11 | 0.03 | 0.26 | | |

[a]no is reference category.

[b]no is reference category.

[c]Two categories "less than once a week" and "at least once a week", where "less than once a week" is the reference category.

[d]Two categories "not living together" and "living together", where "living together" is reference category.

[e]Two categories "urban" and "rural", where urban is reference category.

[f]Two categories "less than 18 years" and "more than 18 years", where less than 18 years is reference category.

*, **, and *** indicate significance at 5%, 1%, and 0.1% levels, respectively.

strengthening the education system especially in the rural region, particularly for the poorest population should be taken as a priority agenda. Similar policy implications were suggested in China [51].

## 5. Strength and limitations

This study has some strength and limitations. One of the major strengths is that we analyzed a nationally representative large cross-sectional data collected through the latest BDHS. Another notable advantage is that we applied the OB decomposition approach which is based on the estimates of coefficient and the mean difference between the groups. Detailed decomposition estimated both explained and unexplained portions of the mean BMI difference, which will help us to determine what difference between the mean of the predictors should be eliminated to remove the inequality of nutrition and which would be remain same even though after changing the difference in the mean. Our study also suffers from some limitations. Firstly, we used only BMI and fail to incorporate other measures to quantify the nutrition of the repro-ductive-aged women due to absence in the BDHS data sets. Nevertheless, waist circumference and waist-to-hip ratio may also be used to measure the nutrition status. Secondly, our results are not comparable to the countries outside of Asia specific region, because we used the Asia specific cut-offs in categorization of BMI. Thirdly, we fail to include some established risk factors such as food habits, life style, physical exercise, smoking, alcohol consumption, which may be the reasons of large unexplained part. Finally, we have analysed only the reproductive-

aged married women which is not generalizable for all reproductive-aged women of Bangladesh.

## 6. Conclusions

Several risk factors for malnutrition including their contribution in explaining the gap between richest and poorest quintiles are estimated for the reproductive-aged women in Bangladesh. It has been found that education of women and their partners, rural-urban place of residence, frequency of TV watching, and current working status of women are the major drivers of the existing nutrition gap between the richest-poorest women. Our study also suggests that to minimize the nutritional gap between wealth categories the difference in mean of those determinants should be minimized. The authorities should initiate some strategies which could reduce the difference in years of education, and healthcare facilities for both urban and rural women. Since increased frequency of TV watching can increase the chance of being overweight and obesity, sitting time in front of TV or electronic devices (smartphone, iphone, ipad) should be avoided as much as possible. Similar decomposition analysis should be performed for all wealth quintiles sequentially. Some important factors namely physical exercise, food habit, visceral adiposity, smoking habit, alcohol consumption, and life styles should be incorporated in further research. Since we have decomposed the nutritional gap based on mean, it fails to decompose the other positional value. So, we suggest other statistical technique such as quantile regression, and median regression to find the coefficients and then use those to find the inequality of the respective positional difference.

## Acknowledgments

We are grateful to the DHS for permitting us to analyse the data. Also, heartfelt gratitude to all the participants of the BDHS.

## Author Contributions

**Conceptualization:** Md. Sohel Rana, Md. Mobarak Hossain Khan.

**Data curation:** Md. Sohel Rana.

**Formal analysis:** Md. Sohel Rana.

**Methodology:** Md. Sohel Rana.

**Software:** Md. Sohel Rana.

**Visualization:** Md. Sohel Rana.

**Writing – original draft:** Md. Sohel Rana.

**Writing – review & editing:** Md. Sohel Rana, Md. Mobarak Hossain Khan.

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
