## [Decision Letter · Decision Letter 0]

20 Apr 2022

PONE-D-21-29922Contribution of Sociodemographic Determinants in Explaining the Nutritional Gap Between the Richest-Poorest Women of Bangladesh: A Decomposition ApproachPLOS ONE

Dear Dr. Rana,

Thank you for submitting your manuscript to PLOS ONE. After careful consideration, we feel that it has merit but does not fully meet PLOS ONE’s publication criteria as it currently stands. Therefore, we invite you to submit a revised version of the manuscript that addresses the points raised during the review process.

We look forward to receiving your revised manuscript.

Kind regards,

Jing Tian

Academic Editor

PLOS ONE

Journal Requirements:

Reviewers' comments:

Reviewer's Responses to Questions

**Comments to the Author**

1. Is the manuscript technically sound, and do the data support the conclusions?

Reviewer #1: Yes

Reviewer #2: Partly

2. Has the statistical analysis been performed appropriately and rigorously? 

Reviewer #1: Yes

Reviewer #2: Yes

3. Have the authors made all data underlying the findings in their manuscript fully available?

Reviewer #1: Yes

Reviewer #2: Yes

4. Is the manuscript presented in an intelligible fashion and written in standard English?

Reviewer #1: Yes

Reviewer #2: No

5. Review Comments to the Author

Reviewer #1: This is an interesting piece of research. Inequality exist everywhere in Bangladesh which slows down the progress in every aspect in the country. This research in important to address the inequality in the context of nutritional status which helps to revise the existence policy. This article has been well written. However, I have some suggestions which helps to upgrade the concept of the article.

Overall the paper needs English proofreading.

In the first para of introduction, “A systematic review of 128 studies, which included the period from 1969 and September 30, 2017, shows that the pooled…” this sentence should not be started like that way. Simply state the prevalence and time of the research.

Introduction section is too long to read and understand. Please narrow down it by one and half page with relevant information.

In the section, results of Table 2, please avoid “We see that” this king of wording.

In discussion section, justify the sentence with ref “The possible reasons behind this matter are that educated peoples are habituated on sedentary life styles, and most of them are doing table-chair oriented work (i.e. passing more time in sitting).

Please add some more policy implications that were successful in others developing countries.

Reviewer #2: Overall comment: The applicability of OB decomposition method over other regression-based methods should be discussed. On many occasions the authors have used the term 'association' inappropriately. The authors tried to demystify the unequal distribution of BMI between the wealthiest and poorest women. But there is a huge overlap in the distribution of BMI between the two groups and the authors didn’t discuss the limitation of OB decomposition techniques in that context. There are many grammatical issues that need to be taken care of; discussion section is full of such errors.

Specific comments:

Abstract:

• Gap in BMI and nutritional gap are not same or similar. BMI is one of the indicators of nutritional status.

Background:

• The study was not done in context of DBM. Hence, this section is overgeneralised and redundant.

• Suffering from overweight or obese: suffering from overweight or obesity

• For instance, undernutrition (underweight, wasting, stunting) can cause communicable diseases: Explain briefly with reference.

• More specifically, underweight is associated with preterm birth and low birth weight [10, 11], malnourished children: Maternal underweight? Please revise carefully to remove all such ambiguities.

Methods:

• The continuous measure of BMI is the ultimate outcome variable and that was used to find the contribution of the predictors. Then why did the authors discuss so much about different BMI cut-offs. Needs clarification.

• What is the benefit of using OB decomposition over regression models? Please clarify for the readers.

• Model selection: What was the statistic the authors used for forward selection? Please mention and clarify. What was the null model? In section 2.3 authors mentioned that they used contemporary literature review, availability of data and appropriateness techniques for selecting the predictors.

Figures:

• Change the title to- “Distribution of BMI according to wealth status”. Association means something different, and the figure is not reporting that.

Tables

• Table 1. Take the ‘%(N)’ row beneath the ‘BMI [Mean (SE)]’ row.

• Table 2. The authors presented the proportion of different variables with the corresponding 95% CIs as per the wealth status. They did not report the association. Please consider reviewing the title.

• Table 3. Participants were grouped based on their wealth status. Then what is role of wealth as a predictor variable? Explain the pooled coefficient of the variable “Wealth”. Interpretation of the terms such as intercept, predicted BMI, total predicted gap is warranted.

Results:

• The unexplained part seems to be casting more impact than the explained part of the OB decomposition results. Please explain this and state the limitations related to this unexplained part.

Discussion:

• From the Kernel density plot we see a huge overlap between the groups. How did the regression model overcome that? How to explain the results in context of such huge overlap.

• “Similarly, mean BMI of the poorest women depicted that they are belonging to normal weight category.”- then why do the authors need to do this decomposition analysis? Just doing a comparison between the overweight/obese and normal weight women would be enough.

• Discussion section should be written focusing on the findings of the study. But here the authors discussed some terms such as physically vigorous situation, burn excess fat of the body, sitting time, contaminated air etc which they didn’t tested as predictors.

• “However, generally we see the women who spend more time in watching TV or smart phone, does very little or no physical movements.”- very strong statement. Do we have supporting data on this?

6. PLOS authors have the option to publish the peer review history of their article (what does this mean?). If published, this will include your full peer review and any attached files.

Reviewer #1: **Yes: **Mohammad Rocky Khan Chowdhury

Reviewer #2: No

---

## [Author Response · Author response to Decision Letter 0]

3 Jun 2022

Dear reviewer's we have uploaded a file named 'Response to Reviewers' where we tried to address your all comments and our explanations. We hope you will satisfy.

---

## [Decision Letter · Decision Letter 1]

17 Aug 2022

Contribution of Sociodemographic Determinants in Explaining the Nutritional Gap Between the Richest-Poorest Women of Bangladesh: A Decomposition Approach

PONE-D-21-29922R1

Dear Dr. Rana,

We’re pleased to inform you that your manuscript has been judged scientifically suitable for publication and will be formally accepted for publication once it meets all outstanding technical requirements.

Kind regards,

Jayanta Kumar Bora, PhD

Academic Editor

PLOS ONE

Additional Editor Comments (optional):

Reviewers' comments:

Reviewer's Responses to Questions

**Comments to the Author**

1. If the authors have adequately addressed your comments raised in a previous round of review and you feel that this manuscript is now acceptable for publication, you may indicate that here to bypass the “Comments to the Author” section, enter your conflict of interest statement in the “Confidential to Editor” section, and submit your "Accept" recommendation.

Reviewer #1: All comments have been addressed

Reviewer #2: All comments have been addressed

2. Is the manuscript technically sound, and do the data support the conclusions?

Reviewer #1: Yes

Reviewer #2: Yes

3. Has the statistical analysis been performed appropriately and rigorously? 

Reviewer #1: Yes

Reviewer #2: Yes

4. Have the authors made all data underlying the findings in their manuscript fully available?

Reviewer #1: Yes

Reviewer #2: Yes

5. Is the manuscript presented in an intelligible fashion and written in standard English?

Reviewer #1: Yes

Reviewer #2: Yes

6. Review Comments to the Author

Reviewer #1: I would like to thank the authors for their efforts in revising the paper. They did wonderfully addressed all of my comments

Reviewer #2: (No Response)

7. PLOS authors have the option to publish the peer review history of their article (what does this mean?). If published, this will include your full peer review and any attached files.

Reviewer #1: **Yes: **Mohammad Rocky Khan Chowdhury

Reviewer #2: No

---

## [Editor Report · Acceptance letter]

23 Aug 2022

PONE-D-21-29922R1 

Contribution of Sociodemographic Determinants in Explaining the Nutritional Gap Between the Richest-Poorest Women of Bangladesh: A Decomposition Approach 

Dear Dr. Rana:

I'm pleased to inform you that your manuscript has been deemed suitable for publication in PLOS ONE. Congratulations! Your manuscript is now with our production department. 

Kind regards, 

on behalf of

Dr. Jayanta Kumar Bora 

Academic Editor

PLOS ONE